# A Comparative Oncology Drug Discovery Pipeline to Identify and Validate New Treatments for Osteosarcoma

**DOI:** 10.3390/cancers12113335

**Published:** 2020-11-11

**Authors:** Jason A. Somarelli, Gabrielle Rupprecht, Erdem Altunel, Etienne M. Flamant, Sneha Rao, Dharshan Sivaraj, Alexander L. Lazarides, Sarah M. Hoskinson, Maya U. Sheth, Serene Cheng, So Young Kim, Kathryn E. Ware, Anika Agarwal, Mark M. Cullen, Laura E. Selmic, Jeffrey I. Everitt, Shannon J. McCall, Cindy Eward, William C. Eward, David S. Hsu

**Affiliations:** 1Department of Medicine, Duke University Medical Center, Durham, NC 27710, USA; gabrielle.rupprecht@duke.edu (G.R.); erdemaltunel90@gmail.com (E.A.); etienne.flamant@duke.edu (E.M.F.); ds311@stanford.edu (D.S.); maya.sheth@duke.edu (M.U.S.); serene.cheng@duke.edu (S.C.); kathryn.ware@duke.edu (K.E.W.); anika.agarwal@duke.edu (A.A.); shiaowen.hsu@duke.edu (D.S.H.); 2Duke Cancer Institute, Durham, NC 27710, USA; jeffrey.everitt@duke.edu (J.I.E.); Shannon.mccall@duke.edu (S.J.M.); william.eward@duke.edu (W.C.E.); 3Department of Orthopaedics, Duke University Medical Center, Durham, NC 27710, USA; sneha.rao@duke.edu (S.R.); alexander.lazarides@duke.edu (A.L.L.); sarah.hoskinson@duke.edu (S.M.H.); mark.cullen@duke.edu (M.M.C.); 4Department of Molecular Genetics and Microbiology, Duke University Medical Center, Durham, NC 27710, USA; soyoung.kim@duke.edu; 5College of Veterinary Medicine, The Ohio State University, Columbus, OH 43210, USA; selmic.1@osu.edu; 6Department of Pathology, Duke University Medical Center, Durham, NC 27710, USA; 7Surgery Service, Triangle Veterinary Referral Hospital, Durham, NC 27710, USA; cindyewarddvm@tvrhdurham.com

**Keywords:** comparative oncology, patient-derived xenografts, precision medicine, CRM1, proteasome, sarcoma

## Abstract

**Simple Summary:**

Osteosarcoma is a rare bone cancer that occurs primarily in children. The discovery of new treatments for osteosarcoma and other rare cancer types has been severely limited by access to patient samples to study these often-complex diseases. Here we capitalize on naturally-occurring cancers in pet dogs to study the biology of these rare cancers. Using living cells from canine and human patients to test thousands of drugs simultaneously, we identify a unique combination of drugs that disrupts protein degradation and protein trafficking in cancer cells. This drug combination represents a promising new treatment to treat both dogs and people with osteosarcoma.

**Abstract:**

Background: Osteosarcoma is a rare but aggressive bone cancer that occurs primarily in children. Like other rare cancers, treatment advances for osteosarcoma have stagnated, with little improvement in survival for the past several decades. Developing new treatments has been hampered by extensive genomic heterogeneity and limited access to patient samples to study the biology of this complex disease. Methods: To overcome these barriers, we combined the power of comparative oncology with patient-derived models of cancer and high-throughput chemical screens in a cross-species drug discovery pipeline. Results: Coupling in vitro high-throughput drug screens on low-passage and established cell lines with in vivo validation in patient-derived xenografts we identify the proteasome and CRM1 nuclear export pathways as therapeutic sensitivities in osteosarcoma, with dual inhibition of these pathways inducing synergistic cytotoxicity. Conclusions: These collective efforts provide an experimental framework and set of new tools for osteosarcoma and other rare cancers to identify and study new therapeutic vulnerabilities.

## 1. Introduction

Osteosarcoma, the most common primary bone cancer, exemplifies the progress that needs to be made in the approach to discovering new therapies. As the third most common cancer of childhood, osteosarcoma is disproportionately lethal, and patients with advanced or metastatic disease have limited treatment options [1]. Due to the low incidence of osteosarcoma and the extensive genetic heterogeneity [2], finding common genetic drivers and common pathways of relevance remains difficult, and the exact etiology remains unknown. Because of these features, progress in identifying new therapies has been slow, and decades of research have brought almost no improvement in patient survival rates [3,4]. Even for those patients who survive, both their life expectancy and quality of life are negatively impacted by the current treatment regimen [5]. For all of these reasons, there is a persistently unmet need to develop new therapies for this deadly disease.

The story of osteosarcoma is the story of many cancers, especially uncommon ones: Alternative therapeutic approaches are urgently needed, but the path forward is not clear. How can we identify, design, and test new molecular therapies in a disease that is both rare and genetically diverse? Mouse models are a critical tool, but additional translational steps are needed. To complete these additional steps, we are able to look to pet dogs with spontaneous osteosarcoma. While there are approximately 800 cases of human osteosarcoma diagnosed each year, there are at least 30,000 cases of canine osteosarcoma diagnosed each year [6,7]. Treatments in canine and human osteosarcoma patients are identical, and studies have revealed remarkable genomic conservation between canine and human osteosarcoma, with shared molecular alterations in known cancer pathways and shared amplifications of known oncogenes [8,9,10,11]. Canine patients with spontaneous disease–in contrast to genetically-engineered mouse models–offer a high incidence of tumors that are comparable to humans biologically and genetically [7,12], share environmental factors with humans, have an intact immune system, and possess similar clinical presentation including progression, resistance, recurrence, and metastasis. Most importantly, canine osteosarcoma patients have a shorter course of disease than human osteosarcoma patients, which means therapeutic discoveries could be made more quickly with a platform that integrates canine osteosarcoma into our current disease models (reviewed in [13]). Naturally-occurring osteosarcoma in dogs offers an unparalleled opportunity to understand the genomics of the disease, to learn about disease progression, and to trial new investigational drugs that would otherwise take too long to accrue in human studies.

While these common features of canine and human osteosarcoma are attractive, it is also important to clearly understand the differences in the biology of canine and human cancers. For example, while osteosarcomas in canines and humans are quite similar in terms of their clinical presentation, progression, host factors (for instance, individuals larger in size are at a higher risk), and molecular profiles (reviewed in [13,14]), there are also some distinct differences, such as the propensity for dogs to be spayed or neutered, which can confound comparisons to human disease (reviewed in [14]). In addition, while both canines and humans have bimodal age distributions in osteosarcoma incidence, human osteosarcoma is more likely to occur during childhood, while most cases of canine osteosarcoma occur in older dogs (reviewed in [14]). Indeed, human osteosarcoma is a disease of childhood and adolescence while canine osteosarcoma is a disease of middle to advanced age. Yet despite these differences, studying naturally-occurring osteosarcoma in dogs offers several advantages to studying osteosarcoma in humans: a significantly larger number of patients, treatment features which are extremely similar, and a shorter course of disease. This provides an unparalleled opportunity to understand the genomics of the disease, to learn about disease progression, and to trial new investigational drugs that would otherwise take too long to accrue in human studies.

Patient-derived models of cancer, including low-passage cell lines [15], patient-derived organoids [16,17], and patient-derived xenografts (PDXs) [18], are increasingly being used as “standard” preclinical models to identify sensitivities to new candidate therapeutics across cancers. Patient-derived xenografts are also being used to predict drug response [19] and identify novel drug combinations [20,21,22,23]. Organoid models are also now being developed to test response to immunotherapies, as the organoids for several cancer types have been shown to contain infiltrating lymphocytes [24,25]. Combinations of these patient-derived models are currently being explored to develop precision medicine strategies for cancer care [26]. However, translating new discoveries in real time remains a challenge in human patients, in whom disease progression can be slow and whose overall picture can be complicated by a variety of treatments, both for the cancer and for comorbid conditions.

Here we combine the advantages of a comparative oncology approach (e.g., larger numbers of patients, fewer confounding treatment variables, more rapid disease progression) with patient-derived models to develop and refine a cross-species drug discovery pipeline. The pipeline uses patient samples from either pet dogs or humans to generate patient-matched, low-passage cell lines, and PDXs. Cell lines are used to perform high-throughput chemical screens, and top hits are validated in vitro using matched PDX models (Figure 1A). Using this approach, we identified the proteasome and CRM1 nuclear export pathways as therapeutic vulnerabilities for osteosarcoma. Using in vitro and in vivo validations, we show that inhibition of both the proteasome and CRM1 pathways acts synergistically to inhibit osteosarcoma growth. Together, these results demonstrate the utility of our cross-species drug discovery pipeline to identify new targets and strategies to treat osteosarcoma and other rare cancers. 

## 2. Results

### 2.1. Development of a Cross-Species Drug Discovery Pipeline

Osteosarcomas and other rare cancers suffer from a lack of access to patient-derived models for study. We reasoned that increased access to a larger patient population with nearly-identical biology could have significant benefits in identifying actionable pathways in osteosarcoma. To this end, we developed a cross-species pipeline that leverages the increased canine patient population and the extensive biological similarities between humans and pet dogs with naturally-occurring sarcomas. The pipeline uses patient tumor tissue from dogs or humans to create patient-derived xenografts (PDXs) that are grown and passaged in immunocompromised mice (Figure 1A). These PDXs are used to create matched, low-passage, patient-derived cell lines. The cell lines are applied to high-throughput screens to identify candidate therapies, and top candidates are validated in vivo using patient-matched PDXs (Figure 1A) to identify therapeutic vulnerabilities that are shared across species. To date, we have created a total of 9 human (Figure 1B) and 20 canine sarcoma PDXs (Figure 1C). 

### 2.2. The Cross-Species Pipeline Identified Proteasome and CRM1 Inhibition as Novel Treatments for Osteosarcoma

We applied our pipeline to reveal novel therapeutic vulnerabilities at both the individual patient level and, more importantly, shared across osteosarcomas. To do this, we created matched cell lines from both dog (D418) and human (17-3X) PDXs. Both PDXs were confirmed as osteosarcoma by histopathology (Figure 2A,B). Using the PDX tissue, we created clonally-purified cell lines. The 17-3X human osteosarcoma cell line exhibits a more spindle-shaped morphology while the D418 dog osteosarcoma cell line displays a rounded, cobblestone-like morphology (Figure 2A,B, right panels). Clonally-derived cell lines were confirmed to be free of detectable mouse fibroblasts using species-specific PCR assays designed to detect human, dog, and mouse DNA (Figure 2C,D). The 17-3X cell line has an estimated doubling time of approximately 40 h (Figure 2E) while the D418 line grows more rapidly, with an estimated doubling time of approximately 21 h (Figure 2F). The 17-3X and D418 low-passage cell lines were combined with a panel of seven additional established osteosarcoma cell lines from both humans and dogs to perform high-throughput screens using 119 FDA-approved small molecule oncology drugs. These screens revealed several trends: (1) cell line-specific variation in responses were observed across the panel, with some cell lines more broadly resistant to drugs in the screen, such as U2OS and D17, and other lines more sensitive, such as 17-3X and Abrams (Figure 3A); (2) hierarchical clustering of the nine cell lines using the drug responses for each cell line revealed species-specific clustering of the cell lines, with distinct human and dog clades formed based on response of each cell line to the entire panel of drugs (Figure 3A); (3) while cell lines clustered by species, there was overall consistency in the average percent killing across all cell lines, particularly among the top hits (Figure 3B). Together, these results suggest that although both individual and species-specific responses exist across osteosarcomas, there are consistent responses to the most effective agents. Importantly, among the top hits were standard-of-care agents such as anthracyclines (e.g., doxorubicin, daunorubicin, idarubicin, epirubicin) and methotrexate, which consistently killed all cell lines, while others showed wider variation (e.g., etoposide) and limited efficacy (e.g., platinum-based chemotherapy) (Figure 3C). These analyses indicate that this screening approach is capable of identifying relevant therapies for osteosarcoma. 

Unlike standard-of-care therapies, analysis of small molecule sensitivities at the individual cell line level revealed heterogeneous responses across the panel of nine cell lines (Figure 3D). Most drugs, such as belinostat (an HDAC inhibitor) or ponatinib (a multi-tyrosine kinase inhibitor), were efficacious across more than one of the nine cell lines (Figure 3D). These analyses reveal that substantial heterogeneity in drug response exists across osteosarcomas. 

Given the extensive heterogeneity of response, we sought to identify novel compounds with efficacy across the entire panel of cell lines. Interestingly, the two most efficacious inhibitors, with an average of >95% killing for both inhibitors across all nine lines, were compounds that target the proteasome pathway (Figure 3E). Together, our results indicate that both human and canine osteosarcomas display heterogeneity in drug response across cell lines, with convergence on the proteasome pathway as a novel target to treat osteosarcoma.

To better understand the landscape of therapeutic vulnerabilities in osteosarcoma, we next performed high-throughput chemical screens in D418 and 17-3X patient-derived lines using 2100 bioactive compounds. This compound library is annotated by both target and pathway, enabling both protein- and pathway-level interrogation of chemical sensitivities. D418 and 17-3X cells displayed similar sensitivity profiles, with just 11.9% and 8.7% of compounds inducing ≥50% killing in D418 and 17-3X, respectively (Figure 4A,B; Appendix A). Consistent with the results from the 119 compound screens, responses of D418 and 17-3X to all 2100 compounds were correlated (R^2^ = 0.54; *p* < 0.0001) (Figure 4C). Also consistent with the previous screens, each cell line displayed sensitivity to a subset of agents (Figure 4D). For example, D418 displayed sensitivity to multiple MEK and FAK inhibitors while 17-3X was uniquely sensitive to Chk inhibitors (Figure 4D). In addition to the unique sensitivities, both cell lines showed common sensitivity to standard-of-care anthracyclines and a number of novel agents (Figure 4E). These agents included the zinc pyrithione, the active ingredient in dandruff shampoo, the pan-selective Jumonji histone demethylase inhibitor, JIB-04, an NF-kB inhibitor (WS3), and two CDK inhibitors (alvocidib and SNS-032) (Figure 4E). Given that almost all small molecule inhibitors have multiple targets, we focused on targets for which at least three drugs showed >50% killing. We reasoned that filtering by drug targets with multiple hits in the screen would identify the most high-confidence drug targets for downstream validation. From these analyses, we identified the CRM1 nuclear export and proteasome pathways as the top candidate targets (Figure 4F). A total of 3 of 4 CRM1 inhibitors and 9 of 11 proteasome inhibitors showed >50% killing in both D418 and 17-3X cell lines (Figure 4G,H). Consistent with these analyses, both the CRM1 inhibitor, verdinexor, and the proteasome inhibitor, bortezomib, showed dose-dependent inhibition of 143B and 17-3X human osteosarcomas (Figure 4I,J) and D418 and D17 canine osteosarcomas (Figure 4K,L), pinpointing the CRM1 and proteasome pathways as lead candidates for in vivo validation.

### 2.3. In Vivo Validation of Proteasome and CRM1 Pathway Inhibitors to Treat Osteosarcoma

Our in vitro small molecule screens pinpointed the proteasome and CRM1 nuclear export pathways as two promising therapeutic vulnerabilities for osteosarcoma. Consistent with this, CRM1 protein is highly expressed (Appendix A) and localized within the nucleus of osteosarcoma cells (Appendix A). Moreover, elevated CRM1 expression is prognostic for poorer metastasis-free and overall survival in human osteosarcoma (Appendix A). We validated the therapeutic efficacy of proteasome and CRM1 inhibition in D418 PDXs and showed that both CRM1 inhibition and proteasome inhibition significantly reduced tumor growth (Figure 5A). CRM1 inhibition also significantly reduced 17-3X PDX tumor growth, while the proteasome inhibitor, bortezomib, had no effect as a single agent (Figure 5B). Mouse weights remained unchanged during the course of the treatment (Appendix A). Based on the positive results for CRM1 inhibitors in two PDXs, we further verified the efficacy of CRM1 inhibition in two additional PDXs, D071 and D075 (Figure 5C,D).

### 2.4. Combined Proteasome and CRM1 Pathway Inhibition Act Synergistically to Prevent Osteosarcoma Growth

Our cross-species pipeline pinpointed both proteasome and CRM1 pathways as promising single-agent therapies to treat osteosarcoma. However, although CRM1 inhibition was effective across all four of the PDXs tested, the proteasome inhibitor, bortezomib, was only capable of inhibiting growth in one of the two treated PDXs when used as a single agent. Interestingly, these pathways are being targeted with combination therapy for synergistic benefit in several cancer types, including multiple myeloma [27], colorectal cancer [28], and fibrosarcoma [29]. Based on these studies and our results, we hypothesized that combined proteasome and CRM1 blockade would have synergistic benefit for osteosarcoma. Consistent with this hypothesis, the combination of proteasome and CRM1 inhibition induced synergistic cell death for both D418 and 17-3X cells (Figure 5E,F), suggesting that this combination may represent a rational strategy to treat osteosarcoma. 

## 3. Discussion

In the present study, we sought to combine the utility of comparative oncology with patient-derived models of cancer and high-throughput small molecule screens to create a cross-species, personalized medicine pipeline. We applied this pipeline to osteosarcoma, a painful bone cancer that occurs predominantly in adolescents and for which almost no treatment progress has been made in nearly four decades. Using our pipeline, we identified therapeutic vulnerabilities for osteosarcoma that are patient-specific, common across patients, and common across species. Among the common therapeutic sensitivities identified were the proteasome and CRM1 nuclear export pathways. Both of these pathways have been explored in depth as potential cancer monotherapies for a range of cancer types [30,31], and both of these therapies are currently approved by the U.S. Food and Drug Administration for the treatment of multiple myeloma [32,33]. 

The proteasome is a multi-subunit complex that degrades misfolded, damaged, or unused proteins. The proteasome regulates the turnover of thousands of proteins in the cell [34], and proteasome inhibition creates an imbalance in the levels of misfolded proteins, leading to induction of the unfolded protein response, cellular stress, and apoptosis [35]. The CRM1 nuclear export pathway transports proteins through the nuclear envelope to the cytoplasm [36]. Like the proteasome, the nuclear export pathway is responsible for regulating thousands of proteins in the cell [37]. Despite the fact that CRM1 is a critical mediator of protein localization for normal cells, there is support for targeting CRM1 in osteosarcoma. Our data show that CRM1 is upregulated in osteosarcoma cells as compared to normal osteoblasts (Appendix A). In addition, in vivo treatment of mice bearing D418 and 17-3X PDXs with the CRM1 inhibitor, verdinexor, had no effect on mouse weight (Appendix A). Third, CRM1 inhibitors are currently being used to treat relapsed, refractory multiple myeloma [38]. Together, our results and current, ongoing clinical trials suggest a therapeutic window exists to rationally utilize CRM1 inhibitors to treat osteosarcoma. 

Interestingly, the proteasome and CRM1 export pathways are functionally linked. Proteasome inhibition in colorectal cancer cells induces CRM1-dependent nuclear export of ubiquitinated proteins, and inhibition of CRM1 prevents this export, leading to cell cycle arrest and apoptosis [28]. In addition, CRM1 inhibition re-sensitizes chemo-resistant myeloma cells to proteasome inhibition [39]. Inhibiting these inter-dependent pathways synergistically inhibits multiple cancers [27,28,29]. Consistent with observations in other cancers, our data support the investigation of proteasome and CRM1 pathway inhibitors for osteosarcoma. It is worth noting we observed more consistent evidence for the use of CRM1 inhibitors, as bortezomib was not effective in one of the two matched cell line-PDX models studied. The reasons for the discrepancy between our in vitro and in vivo data on bortezomib are likely due to numerous factors, such as cross-talk with tumor-associated stromal cells [40]; however, despite this discrepancy we observed that combined proteasome and CRM1 pathway inhibition led to synergistic cytotoxicity in vitro, providing further evidence for the efficacy of dual proteasome and CRM1 pathway inhibition in treating osteosarcoma. Future canine clinical trials will provide important insights into the tolerability and efficacy of this combination therapy.

Both of these targets/pathways are currently being explored as treatments for pediatric cancers, with promising tolerability and efficacy. In a phase I tolerability study, the CRM1 inhibitor selinexor was tolerable at doses as high as 55 mg/m^2^ in combination with fludarabine and cytarabine in pediatric patients with relapsed or refractory leukemia [41]. Likewise, the addition of bortezomib in combination with reinduction therapy led to remissions in 8/10 patients, with grade 3 infections in 4/10 patients [42]. Unlike selinexor and bortezomib, which appear tolerable when combined with other chemotherapeutic agents, the standard-of-care treatment regimen for osteosarcoma is a combination of cytotoxic methotrexate, adriamycin (doxorubicin), and cisplatin, which is associated with substantial treatment-derived morbidity [43,44,45]. Side effects from these agents can delay treatment regimens in pediatric and adolescent patients, and more severe complications, such as anthracycline-induced cardiomyopathy, greatly increase subsequent morbidity [46,47]. Indeed, even osteosarcoma survivors do not have a normal life expectancy due to the morbidity associated with their therapy during childhood.

The establishment of a cross-species drug discovery pipeline provides a robust platform to identify and validate potential new therapies and offers several advantages. First, it capitalizes on the expanded canine patient population with spontaneous disease, thereby substantially improving access to patient samples for the development of patient-derived models of cancer. This is particularly useful for studying rare cancers. Indeed, we established more than double the number of PDXs from canines than from humans. Second, combining low passage and established cell lines with high-throughput chemical screens enables interrogation of hundreds to thousands of compounds simultaneously, pinpointing both patient-specific and population-level therapeutic vulnerabilities. Third, the use of chemical screens with complete target and pathway annotation allows for identification of single agent- and target/pathways-level drug sensitivities. Fourth, the use of matched cell lines and PDXs from the same patient provides a robust system to validate top candidates. While the in vivo studies are a critical validation step in the pipeline, these studies are costly and time consuming. Future iterations of the pipeline that exploit continued improvement in patient-derived models, such as patient-derived organoids [17,48,49], are sure to improve the speed and cost-effectiveness of the pipeline and will further enable rapid translation of lead candidates into clinical practice [21]. Fifth, and perhaps most critically, the ability to test these candidates in veterinary clinical trials “closes the loop” of drug discovery (Figure 1A) and enables testing of novel therapeutic strategies at a fraction of the cost and time necessary for human trials (reviewed in [13]). Using this comparative oncology pipeline we identified two promising inhibitor classes, targeting the proteasome and CRM1 nuclear export pathways, with in vitro efficacy as single agents and in vitro synergy in combination. Together, our data suggest that future studies should be focused on testing the in vivo efficacy of combined targeting of the proteasome and CRM1 pathways in osteosarcoma. Canine clinical trials may pave the way for subsequent human clinical trials aimed at prolonging the lives of osteosarcoma patients.

## 4. Materials and Methods

### 4.1. Generation of the Patient-Derived Xenograft Models

Canine sarcoma tumor tissue samples were collected from University of Illinois at Urbana Collage of Veterinary Medicine (Urbana, IL, USA) and Triangle Veterinary Referral Hospital (Durham, NC, USA) under Institutional Animal Care and Use committee (IACUC)-approved protocols. Human sarcoma tumor tissue samples were collected under a Duke IRB approved protocol (Pro00002435). All patients provided written informed consent to participate in the study. PDX models of the human and canine sarcoma were generated as described previously [21] and all mouse experiments were performed in accordance with the animal guidelines and with Duke University IACUC approval (A063-18-03). To develop PDXs, human and canine tumor tissue samples were washed in phosphate buffered saline (PBS), dissected into small pieces (<2 mm), and injected into the flanks of 8–10-week-old JAX NOD.CB17-PrkdcSCID-J mice obtained from the Duke University Rodent Genetic and Breeding Core.

### 4.2. Patient-Derived Cell Line Generation, Authentication and Detection of Mouse Cell Contamination

Cell lines were generated from human and canine tumor samples. After washing in phosphate buffered saline (PBS), small pieces (<2 mm) of tumor tissue were mechanically homogenized. The homogenized tissue was then suspended in cell growth media and cultured in 12 well plates with DMEM + 10% FBS + 1% Penicillin/Streptomycin. To isolate tumor cells, growing colonies of cells were isolated by trypsinization using O rings and cultured in new 12 well plates. This process was repeated until a colony of cells that resembled pure tumor cells in morphology was established.

For the cell lines generated from PDX, mouse cell contamination of the PDX cell lines were detected by PCR using human, canine, and mouse specific primers. Because mouse primers easily cross-react with canine and human gDNA, two different mouse primer sets were used. For human cell lines, human reverse (5′ TCC AGG TTT ATG GAG GGT TC), human forward (5′ TAG ACA TCG TAC TAC ACG ACA CG), mouse reverse (5′ CCC AAA GAA TCA GAA CAG ATG C) and mouse forward (5′ ATT ACA GCC GTA CTG CTC CTA T); for canine cell lines, canine reverse (5′ GTA AAG GCT GCC TGA GGA TAA G), canine forward (5′GGT CCA GGG AAG ATC AGA AAT G), mouse reverse (AGG TGT CAC CAG GAC AAA TG) and mouse forward (CTG CTT CGA GCC ATA GAA CTA A) primer sets were used.

All PDX and patient-derived cell lines were authenticated using the Duke University DNA Analysis Facility cell line authentication service by analyzing DNA samples from each individual cell line for polymorphic short tandem repeat markers using the GenePrint 10 kit from Promega (Madison, WI, USA).

### 4.3. In Vitro Studies and High-Throughput Drug Screen

All human (143B, MG63, SAOS, U2OS, 17-3X) and canine (Abrams, Moresco, D17, D418) osteosarcoma cell lines were cultured in DMEM + 10% FBS + 1% Penicillin/Streptomycin. The NIH Approved Oncology set (119 compounds) and Selleck Bioactives collection (2100 compounds) were screened in the Duke Functional Genomics Shared Resource as described previously [21,50]. Briefly, compounds were first stamped in triplicate into 384 well plates at a final concentration of 1 μM using an Echo Acoustic Dispenser (Labcyte, Indianapolis, IN, USA). Cells and media were then dispensed into plates using a WellMate (Thermo Fisher, Waltham, MA, USA) at a density of 2000 cells/well for each cell line. CellTiter-Glo (Promega, Madison, WI, USA) viability assays were performed after incubation of cells with compounds for 72 h and luminescence was read using a Clariostar plate reader (BMG, Berlin, Germany). Top drug targets as identified by the high-throughput drug screens, Bortezomib (PS-341), Verdinexor (KPT-335) and 17-DMAG (Alvespimycin) HCl were purchased from Selleck Chemicals (Houston, TX, USA) and were solubilized in DMSO at 10 mM concentration to use for in vitro IC50 studies. Dose-response curves were generated with four-fold serial dilutions from a starting concentration of 20 μM.

### 4.4. In Vivo Drug Sensitivity Validation

To validate in vitro drug screen results in vivo, 150 μL homogenized PDX tissue-PBS suspensions at 150 mg/mL concentration were injected subcutaneously into the right flanks of the 8 weeks old JAX NOD.CB17-PrkdcSCID-J mice. When the tumor volumes reached 100 mm^3^, mice were randomized (*n* = 5 mice for each treatment group and *n* = 5 for the control group) and 1 mg/kg bortezomib, 25 mg/kg alvespimycin and 5 mg/kg verdinexor intraperitoneal injections were initiated twice a week. Tumor volumes were measured three times a week using calipers, and $\frac{\left(Length\;\times \;Width^{2}\right)}{2}$ was used to calculate the tumor size. Mice were sacrificed on day 18 or if the tumor volume reached 1500 mm^3^.

### 4.5. Data Analysis and Statistics

JMP from SAS software (Cary, NC, USA) was used for the high-throughput drug screen data analysis. Analysis of Means was used to identify the top drug candidates from the 119-compound drug screen and the 2100-compound screen. Tumor volumes were recorded in GraphPad Prism 6 software (La Jolla, CA, USA). Two-way ANOVA analysis was used to compare the tumor volumes among the control and treatment groups. Kaplan Meier curves for CRM1 expression were generated using the R2 genomics software (https://hgserver1.amc.nl/cgi-bin/r2/main.cgi) with the “Kaplan Meier by gene expression” tool and the cutoff/modus set to “scan”.

## 5. Conclusions

We demonstrate here the utility of combining patient-derived models of cancer with comparative oncology approaches to discover potential new therapies for osteosarcoma. The increased access to patient samples with biologically-similar disease not only means we can make discoveries in the lab faster, but we can also validate lead candidates in veterinary clinical trials at a fraction of the time and cost. Increasing collaboration between veterinarians, cancer researchers, and human physicians is needed to apply this pipeline and identify new therapies to treat osteosarcoma and other cancers across species.

## Figures and Tables

**Figure 1 cancers-12-03335-f001:**
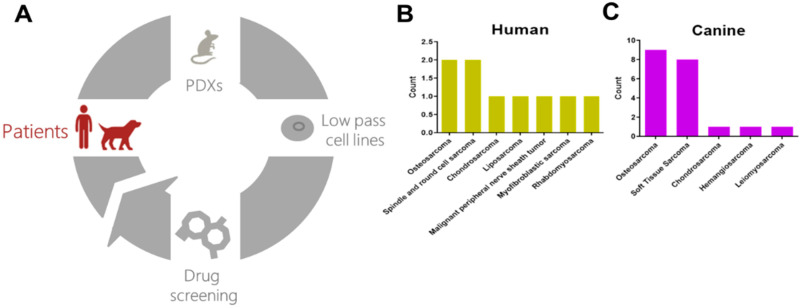
A cross-species personalized medicine pipeline using patient-derived models of cancer. (**A**) The pipeline uses tumor samples from human and canine patients to establish matched patient-derived xenografts and low-passage cell lines. The cell lines are used in high-throughput drug screens, and results from the screen are validated in matched patient-derived xenografts. (**B**) A summary of human (top) and (**C**) dog (bottom) samples obtained and number of patient-derived xenografts created.

**Figure 2 cancers-12-03335-f002:**
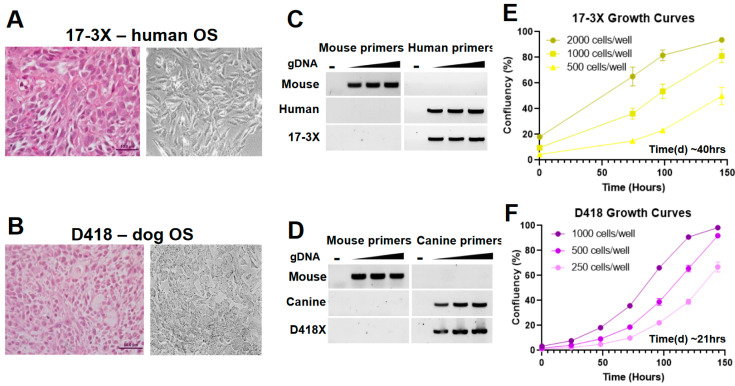
Cross-species analysis of drug activity reveals remarkable similarity in response. (**A**) Establishment of matched patient-derived xenografts and cell lines from human (17-3X) and (**B**) dog (D418) osteosarcomas. (**C**,**D**) Species-specific PCRs are used to verify the cell lines are purified cancer cell lines devoid of mouse fibroblast contamination. (**E**) The estimated doubling times for the 17-3X and D418 cell lines are approximately 40 and (**F**) 21 h, respectively.

**Figure 3 cancers-12-03335-f003:**
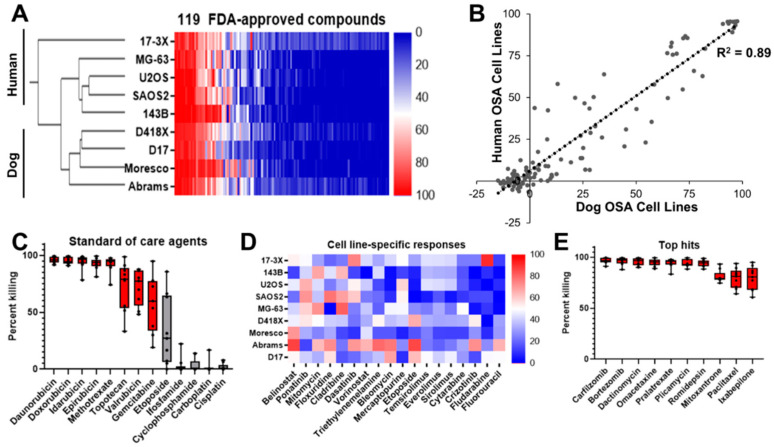
Cross-species analysis of osteosarcoma drug response reveals sensitivity to proteasome inhibition. (**A**) A high-throughput screen of 119 oncology compounds across nine osteosarcoma cell lines revealed species-specific clustering by drug response. (**B**) Although both individual and species-specific responses exist across osteosarcomas, there is a strong correlation between dog and human cell lines (R^2^ = 0.89). (**C**) Standard-of-care agents, such as anthracyclines and methotrexate are among the top hits. (**D**) Cell-line specific responses vary widely to targeted agents and other chemotherapeutics. (**E**) Proteasome inhibitors carfilzomib and bortezomib demonstrate efficacy across all nine cell lines.

**Figure 4 cancers-12-03335-f004:**
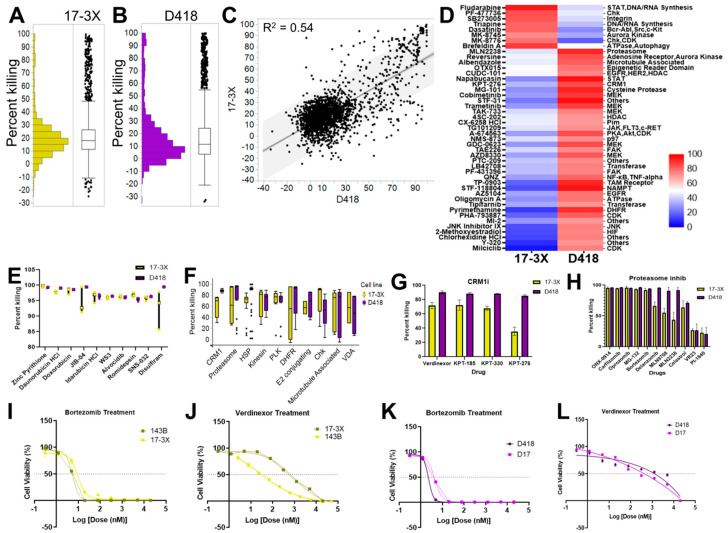
Interrogating the therapeutic landscape of osteosarcoma pinpoints the proteasome and nuclear export pathways as promising therapeutic targets. (**A**) Chemical screens were performed using 2100 compounds in 17-3X and (**B**) D418 low-passage cell lines. (**C**) Drug response was correlated across species (R^2^ = 0.54). (**D**) Cell line-specific sensitivities for 17-3X and D418 cell lines. (**E**) Top drugs, and (**F**) top pathways for both cell lines. (**G**) Cell line-specific response to each of the CRM1 inhibitors and (**H**) proteasome inhibitors. (**I**) IC50 dose response curve for bortezomib and (**J**) verdinexor in 143B and 17-3X human cell lines. (**K**) IC50 dose response curves for bortezomib and (**L**) verdinexor in canine D418 and D17 cell lines.

**Figure 5 cancers-12-03335-f005:**
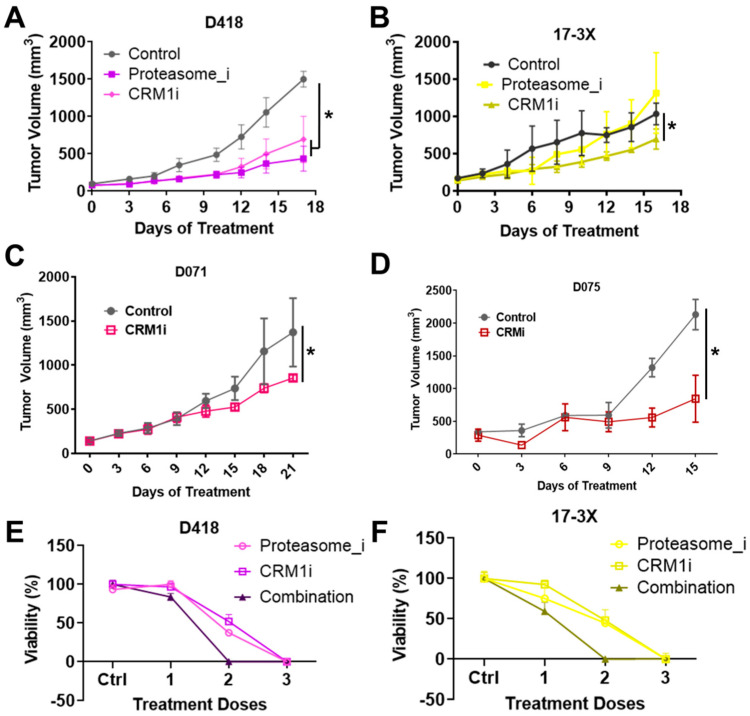
Proteasome and CRM1 nuclear export pathway inhibition reduces osteosarcoma tumor growth and induces synergistic killing of osteosarcomas. (**A**) CRM1 inhibition (verdinexor), but not proteasome inhibition (bortezomib) significantly reduced tumor growth in 17-3X. (**B**) Both CRM1 and proteasome inhibition significantly reduced D418 tumor growth. (**C**,**D**) CRM1 inhibition significantly reduced tumor growth rate of D071 and D075 patient-derived xenografts in vitro. (**E**) Combined CRM1 and proteasome inhibition led to synergistic inhibition of 17-3X and (**F**) D418 cell growth. * (*p* < 0.05).

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
