# Peer review of "A Comparative Oncology Drug Discovery Pipeline to Identify and Validate New Treatments for Osteosarcoma"

_cancers, 2020, doi:10.3390/cancers12113335_

Round 1

Reviewer 1 Report

Sir,

I have recently reviewed the manuscript "A Cross-Species Drug Discovery Pipeline to Identify and Validate New Treatments for Osteosarcoma" submitted to Cancer by Somarelli, Rupprecht & co-workers. 

Mesenchymal tumours represented here by osteosarcoma are indeed rare, yet aggressive diseases worthy of attention. Frequently deadly, frequently devastating, usually poorly predictable because we lack the understanding of the biology of sarcomas. Authors offer a certain comparative approach and assume that this combination might bring a more in-depth insight into human osteosarcoma. Working with animal models (of human diseases) for some years,  I am interested in their biological observations, but I am sceptical in terms of immediate oncologic applicability. 

My criticism is the following: 

  • authors compare canine and human osteosarcoma. I do NOT claim that it is worse than comparison with murine osteosarcoma. I am not anyhow biased here. But it simply remains a comparison of 2 species, nothing more. Therefore I find the headline probably too ambitious. More humble word selection would be more pertinent. 
  • As a dog-lover and vet-related bill payer, I can also clearly understand that the availability of these samples for research is tempting because of usually relatively well-documented disease history and diagnostic/therapeutic options comparable to human medicine. Available does not necessarily mean optimal. I would advise to include a brief paragraph on summarizing specific features of canine osteosarcoma. This is not completely identical with human ones. This issue must be honestly presented as well. 
  • In the most critical part of this manuscript (CRM1 and proteasome inhibition as novel treatments for osteosarcoma)  authors claim"To do this, we created matched cell lines from both dog (D418) and human (17-3X) PDXs. And further (lines 136-138): To better understand the landscape of therapeutic vulnerabilities in osteosarcoma, we next performed high-throughput chemical screens in D418 and 17-3X patient-derived lines using 2,100 bioactive compounds.  I am sorry, but this is just ONE cell line from every species. The general rules given by MDPI written in instructions for authors say the following: research articles using only one cell line for the experiments will not be considered for publication. Because I entirely believe authors that "Cell line-specific responses vary widely to targeted agents and other chemotherapeutics" I must also believe that this design 1 human + 1 canine is not, therefore, robust and scientifically sufficient. 
  • Using 9 different osteosarcoma cell lines, authors observed certain variability cell lines in general terms of toxicity and drug response. Unfortunately, they did not demonstrate any data on normal (non-malignant) cell populations. CRM1 (Chromosomal Maintenance 1, also known as Exportin 1) is the major mammalian export protein that facilitates the transport of large macromolecules including RNA and protein across the nuclear membrane to the cytoplasm. This is very likely that inhibition will also affect normal bone cells.  In recent years, the cancer-cell specific mechanisms are favoured for therapeutic targeting. This aspect is not touched at all in the presented study ...

Sir, in my eyes, this manuscript possesses certain potential. However, I must honestly say that the above-listed controversies do not allow me to support it now. Authors should revise their data more stringently. I also believe that additional experiments are needed to gain more robust data. 

Reviewer 2 Report

In this paper “A cross-species drug discovery pipeline to identify and validate new treatments for osteosarcoma” Somarelli J and collaborators performed an in vitro high-throughput drug screens on low-passage and established cell lines with in vivo validation in patient derived xenografts. They identify the proteasome and CRM1 nuclear export pathways as therapeutic sensitivities in osteosarcoma. The authors provide an experimental framework and set of new tools for osteosarcoma and potentially other rare cancers to identify.

This article was well written. In my opinion this article is suitable for publication but after a minor revision.
Specific comments

1) The authors should specify the dose of treatment for specific inhibitors.

2) Could be very useful to summarize all inhibitors in a table and indicate which inhibitor have been used (e.g. which MEK inhibitors? FAK inhibitors? etc).

3) Fig 5: authors should include description of panel E and F in figure legend

4) The supplementary Fig 1 legend is missing.

5) Suppl Figure 1 C and D: authors should describe how metastasis-free and overall survival in human osteosarcoma have been analysed. Kaplan-Meire plot should be labelled with the group criteria (cut-off value to define the high or low-risk group) and patient number in each group.

Reviewer 3 Report

The manuscript by Somarelli et al. describes a process of drug screening for osteosarcoma in a cross-species drug discovery system utilizing low passage cells established from PDXs both human and canine tumors. The authors identified drugs inhibiting proteasome pathway and CRM1 nuclear export pathway as candidates for osteosarcoma treatment and tested their efficiency both in vitro cell and in vivo models. They provided a valid approach to screen candidate drugs in, so called, a cross-species drug discovery system for rare diseases, but with available orthologous animal ones. Presented analyses with chemical compounds appear to be carried out reasonably and acceptably, which could be followed by colleagues in the relevant area. There are a few points that should be addressed properly to make the authors’ discovery more impactful.

1. It is advised to provide a comparison between the effects of proteasome inhibitors, CRM1-dependent export inhibitors and currently used chemotherapeutics.

2. In lines 271 and 279 of ‘Materials and methods’, the authors mentioned alvespimycin treatment, but the results are not provided. The authors are advised to either include or discuss results with the drug.

3. It is a surprise that bortezomib did not have any effect on in vivo tumor model with human osteosarcoma. Discussion on the probable cause of the discrepancy should be provided.

4. It is highly recommended to examine the effect of combination treatment of bortezomib and verdinexor in vivo.

Reviewer 4 Report

In this article are well described the data and the experiments to validate this method for development of a cross-species drug discovery pipeline to identify new targets and strategies to treat osteosarcoma and other rare cancers.
Have the authors tried to test also the effects of different natural compounds?

Round 2

Reviewer 1 Report

Sir, 

I have studied the rebuttal letter of Dr Somarelli with utmost care earlier today. 

I believe that the revisions performed over last month were somewhat helpful to strengthen the manuscript.  Indeed, I see a positive development in this case. I am grateful for courtesy of provided answers and I can confirm that Authors addressed all issues honestly and the modifications are scientifically sound. 

Regarding my comment Nr. 3 ("the larger screens of 2,100 compounds performed in two separate cell lines"), I still insist on a stringent way of interpretation. These two cell lines should not be of two different species (1 canine + 1 human) because these could represent potentially two (non-identical/non-comparable) pathological machineries (with different pathological mechanisms running behind). It would be definitely more robust to have 2 canine vs 2 human cell lines compared and presented here. I am also conscious of the potentially high financial expenses of doing so.  Also, the authors elaborated on the cross-species sarcoma comparison difficulties- see their response to my Comment 2. This was clarified to potential readers well.  

Therefore, I am rather keen to accept the author´s explanation why this study design is a bit shallow here and how it is compensated elsewhere (the results were consequently validated in vivo using FOUR separate patient-derived xenografts). I believe that this allows a plausible overcoming of this bottleneck.

At this stage, I believe that my decision must be binary. To champion the research of rare diseases (including rare malignancies like osteosarcoma) I believe that I am entitled to support publication of this revised paper.